# Design and Synthesis of a Novel NIR Celecoxib-Based Fluorescent Probe for Cyclooxygenase-2 Targeted Bioimaging in Tumor Cells

**DOI:** 10.3390/molecules25184037

**Published:** 2020-09-04

**Authors:** Xinli Wang, Liye Wang, Lijun Xie, Zuoxu Xie, Li Li, Dinh Bui, Taijun Yin, Song Gao, Ming Hu

**Affiliations:** 1Department of Medical Oncology, Fujian Medical University Union Hospital, Fuzhou 350001, Fujian, China; xinliwang528@outlook.com; 2Department of Pharmacological and Pharmaceutical Sciences, College of Pharmacy, University of 10 Houston, Houston, TX 77204, USA; liyewang0110@gmail.com (L.W.); garrison1225@163.com (L.X.); xiezuoxu@hotmail.com (Z.X.); lilihbum@gmail.com (L.L.); dinhbt1988@gmail.com (D.B.); tyin@central.uh.edu (T.Y.); song.gao@tsu.edu (S.G.); 3Fujian Provincial Key Laboratory of Screening for Novel Microbial Products, Fujian Institute of Microbiology, Fuzhou 350007, Fujian, China

**Keywords:** COX-2, celecoxib, CY-5, probe, cancer

## Abstract

Cyclooxygenase-2 (COX-2) imaging agents are potent tools for early cancer diagnosis. Almost all of the COX2 imaging agents using celecoxib as backbone were chemically modified in the position of N-atom in the sulfonamide group. Herein, a novel COX-2 probe (CCY-5) with high targeting ability and a near-infrared wavelength (achieved by attaching a CY-5 dye on the pyrazole ring of celecoxib using a migration strategy) was evaluated for its ability to probe COX-2 in human cancer cells. CCY-5 is expected to have high binding affinity for COX-2 based on molecular docking and enzyme inhibition assay. Meanwhile, CCY-5 caused stronger fluorescence imaging of COX-2 overexpressing cancer cells (Hela and SCC-9 cells) than that of normal cell lines (RAW 264.7 cells). Lipopolysaccharide (LPS) treated RAW264.7 cells revealed an enhanced fluorescence as LPS was known to induce COX-2 in these cells. In inhibitory studies, a markedly reduced fluorescence intensity was observed in cancer cells, when they were co-treated with a COX-2 inhibitor celecoxib. Therefore, CCY-5 may be a selective bioimaging agent for cancer cells overexpressing COX-2 and could be useful as a good monitoring candidate for effective diagnosis and therapy in cancer treatment.

## 1. Introduction

Cancer remains as the leading cause of death worldwide. Cancer-related deaths can be decreased by 30% among cancer patients who are diagnosed and treated at the early stages [1,2], which highlights the importance of developing novel strategies for early diagnosis. Consequently, great efforts have been made over the last decade in the development of imaging technologies for cancer early diagnosis. Several methods have been used for cancer imaging, including single-photon emission computed tomography (SPECT) and positron emission tomography (PET), computed tomography (CT), radionuclide imaging using single photons and positrons, magnetic resonance imaging (MRI), ultrasonography (US), and optical imaging. However, conventional examination techniques failed to provide enough contrast for sensitive and reliable identification of early tumor diseases [3,4].

To address this issue, fluorescent imaging methods have drawn considerable attention because of their high selectivity and sensitivity, in addition to relatively low cost and noninvasive nature [5,6,7,8]. Fluorescent probes have been used for the early diagnosis through detecting the enzymes which were highly expressed in different types of cancers [9,10,11]. Clinical outcomes showed that COX-2 is an overexpressed enzyme in tumors at all the stages, nor or poorly expressed in normal tissues [12,13,14,15,16]. As such, the detection and imaging of COX-2 with specific fluorescent probes could provide valuable information to enable the early diagnosis of cancer [17,18]. Accordingly, COX-2 fluorescent probes have been emerging as a promising tool for detecting early-stage cancers. In recent years, many COX-2 fluorescent probes have been developed and shown to determine the expression levels of COX-2 in cells and early-stage tumor tissues. Conventionally, most of them are synthesized by conjugating a bulky fluorophore to the drug of interest via an appropriate flexible linker [19,20,21,22,23,24,25,26,27,28,29,30]. Exemplarily, Marnett and colleagues established a series of effective COX-2-targeted fluorescent imaging agents in vitro and in vivo [25,26,27]. Peng and co-workers developed several fluorescent probes targeting COX-2 in cancer cells. They prepared a naphthalene-based two-photon optical probe for real-time bio-imaging of COX-2 in living biosystems [28,29,30]. Notably, Chen recently reviewed the progress of COX-2 fluorescent probes and demonstrated that fluorescence detection and imaging of COX-2 is a vital technology for early diagnosis of cancers [24].

Based on the above observation, although many fluorescent probes targeting COX-2 have been developed [12], inadequate targeting specificity limits their further clinical applications. As such the discovery of COX-2-specific cancer biomarkers is currently still in the early stages. After investigating structurally various celecoxib-derived COX-2 fluorescent probes, it appears to us that all were chemically modified in the position of N-atom in the sulfamide group [19,20,21,30] (Figure 1). Duo to the vital role of this group played in the binding to the COX-2 enzyme, as reported by Monahan [31], we anticipated that if we tried to modify the celecoxib skeleton by migrating fluorophore moiety from the N position to pyrazole ring position as shown in Figure 1, it offered a high probability that a new structural space for obtaining more potent COX-2 probes may exist with potential of increased COX-2 targeting ability.

Prompted by the findings mentioned above, in our study, a novel NIR fluorescent probe celecoxib-CY5 probe (CCY-5) with COX-2 targeting ability, has been designed and synthesized. Celecoxib was chosen as a recognition group for the first time, to the best of our knowledge, linking to a CY5 via a flexible carbon amide linker, constructed a targeted-COX-2 fluorescent probe CCY-5 with excitation (at 648 nm) and emission (at 673 nm). In subsequent studies, the determination of affinity and the inhibitory ability for COX-2 was evaluated by computer verification and inhibition assay. The COX-2 inhibition ability of CCY-5 is comparable with that of compound CMP [20], 11.2 and 14.2-fold decreased respectively when taking celecoxib as the positive control. We demonstrated that CCY-5 could selectively accumulate in cancer cell lines HeLa, SCC-9 as well as RAW 264.7 cell pre-treated with LPS. Through comprehensive verifications, we believe that the probe CCY-5 is capable of specifically detecting and imaging of COX-2 in vitro.

## 2. Materials and Methods

### 2.1. Materials and Instrumentations

All the reagents were used as received without further purification. Column chromatography purifications were performed using silica gel 60 (Merck Millipore, 0.040–0.063 mm) as a stationary phase. Sixty silica gel plates (Merck Millipore 60 F254) were used for analytical thin-layer chromatography. The mass spectra were recorded on an Ion Spec Hi Res ESI Shimadzu LC/MS-2020 mass spectrometer and LC/MS-8050 mass spectrometer (Shimadzu, Kyoto, Japan). The ^1^H and ^13^C-NMR spectra were collected on the JEOL 600 MHz spectrometers (Framingham, MA, USA) using CDCl_3_ and DMSO-*d_6_* with TMS used as an internal reference.

### 2.2. Synthesis

Probe CCY-5 was synthesized as shown in Scheme 1. The scaffold of compound **1** was synthesized using previously reported methods [32]. The reaction of **1** with sodium azido yielded compound **2**, which was transformed to compound **3** via Staudinger reduction, the polarity of compound **3** was increased significantly compared with that of compound **2**. Finally, through the traditional condensation reaction, a CY-5 moiety was successfully conjugated with amine **3** to give the final compound CCY-5. The identity of CCY-5 was verified by HR-MS, ^1^H-NMR, and ^13^C-NMR spectroscopy.

#### 2.2.1. Synthesis of Compound **2**

Compound **1** (3.6 g, 8.9 mmol) was dissolved in DMF (50.0 mL) and the solution was stirred at 55 °C, catalytic quantity KI (20 mg) and NaN_3_ (0.86 g, 13.3 mmol) was added into the solution and further stirred for 1 h. The reaction mixture was diluted with Ethyl acetate and washed thoroughly with water, 1 N HCl aqueous solution, NaHCO_3_ aqueous solution and brine, dried over Na_2_SO_4_, filtered, and concentrated. Silica gel chromatography (Petrol/Ethyl acetate = 1:1) of the crude mixture afforded compound **2** (2.4 g, 73.4%). MS (ESI, *m/z*): 369.0; ^1^H-NMR (600 MHz, DMSO-*d*_6_) δ 7.80 (d, *J* = 8.9 Hz, 2H), 7.44 (s, 2H), 7.41 (d, *J* = 8.9 Hz, 2H), 7.17 (d, *J* = 8.7 Hz, 2H), 7.12 (d, *J* = 8.6 Hz,2H), 6.66 (s, 1H), 4.48 (s, 2H), 2.28 (s, 3H). ^13^C-NMR (151 MHz, DMSO-*d*_6_) δ 149.1, 144.6, 143.3, 142.3, 138.9, 130.0, 129.9, 129.8, 129.1, 128.9, 127.2, 127.0, 125.6, 108.3, 47.5.

#### 2.2.2. Synthesis of Compound **3**

A 100-mL round-bottomed flask was charged with compound **2** (2.4 g, 6.5 mmol) and THF (60.0 mL). To this solution, triphenylphosphine (3.4 g, 13.0 mmol) was added slowly, and the reaction mixture was stirred for 2 h at 60 °C. Water (0.5 mL) was added, and the resulting suspension was stirred for 6 h. The mixture was concentrated under reduced pressure. The crude mixture was purified by flash column chromatography (silica gel, dichloromethane/methanol, 200:1) to afford the desired compound **3** (1.4 g, 65%) as a yellow powder. MS (ESI, *m/z*): 343.1; ^1^H-NMR (600 MHz, DMSO-*d*_6_) δ 7.76 (d, *J* = 8.6 Hz, 2H), 7.36 (d, *J* = 8.6 Hz, 2H), 7.17 (d, *J* = 7.9 Hz, 2H), 7.09 (d, *J* = 8.0 Hz, 2H), 6.57 (s, 1H), 3.70 (s, 2H), 2.27 (s, 3H). ^13^C-NMR (151 MHz, DMSO-*d*_6_) δ 157.0, 143.9, 142.7, 142.7, 138.6, 130.0, 129.9, 129.8, 128.9, 128.8, 128.8, 127.7, 127.1, 125.2, 125.1, 107.7, 21.3.

#### 2.2.3. Synthesis of CCY-5

Compound **3** (50 mg, 0.15 mmol), *N*-(3Dimethylaminopropyl)-*N*′-ethylcarbodiimide hydrochloride (EDCI, 31 mg) and 4-dimethylaminopyridine (DMAP, 2.0 mg) was solved in DCM (5 mL) at 0 °C. After 30 min, CY-5 (76 mg, 0.15 mmol) was added to the solution, followed by stirring at room temperature for overnight. As monitored by TLC, the solvent was removed and was washed with 1N HCl solution, organic layer was dried with anhydrous Na_2_SO_4_. Main product was purified by flash column chromatography (ethyl acetate/dichloromethane,1:50 *v/v*) (56 mg, 48%). ^1^H-NMR (600 MHz, CDCl_3_): δ 7.90–7.78 (m, 2H), 7.74–7.62 (m, 2H), 7.40–7.29 (m, 5H), 7.22–7.15 (m, 4H), 7.10–7.01 (m, 5H), 7.00–6.94 (m, 2H), 6.68 (t, *J* = 11.0 Hz, 1H), 6.42 (s, 1H), 6.17 (d, *J* = 11.5 Hz 2H), 5.93 (s, 2H), 5.29 (s, 1H), 4.61–4.47 (m, 2H), 4.01–3.88 (m, 2H), 3.54 (s, 3H), 3.20–3.08 (m, 2H), 2.42–2.32 (m, 2H), 2.28 (s, 3H), 2.05–1.97 (m, 2H), 1.87–1.75 (m, 5H), 1.68 (s, 6H), 1.66 (s, 6H), 1.60–1.46 (m, 2H), 1.30–1.21 (m, 2H), 0.90–0.79 (m, 2H). ^13^C-NMR (151 MHz, CDCl_3_) δ 173.0, 153.3, 153.1, 152.1, 142.8, 142.4, 142.0, 141.2, 140.9, 140.6, 139.0, 129.5, 129.2, 128.8, 128.7, 128.6, 127.8, 127.5, 126.2, 125.4, 125.3, 125.2, 125.0, 123.3, 122.2, 122.1, 116.7, 110.7, 110.5, 108.1, 103.8, 49.4, 49.2, 44.2, 28.0, 27.9, 27.1, 26.5, 25.2, 23.5, 21.3. HR-MS (ESI): calcd for C_49_H_55_N_6_O_3_S^+^: 807.4051 ([M]^+^), found: 807.4031.

### 2.3. UV/vis and Fluorescence Spectroscopic Methods

All the UV/Vis absorption spectra and fluorescence were recorded on S-3100 and RF-5301PC spectrophotometer (Shimadzu, Kyoto, Japan), respectively. Stock solutions (500 μM) of probes were prepared in DMSO, and then diluted into 10 μM solution of probes in PBS buffer with 2% DMSO. In all experiments, the excitation wavelength was 648 nm and the emission wavelength was 673 nm.

### 2.4. Cell Culture

Human cervical cancer (HeLa), human oral squamous cell carcinoma (SCC-9), mouse monocyte/macrophage (RAW 264.7) cell lines were obtained from the American Type Culture Collection. Hela and RAW 264.7 cells were cultured in Dulbecco’s modified Eagle’s media (Hyclone) supplemented with 10% fetal bovine serum (Hyclone) and 1% penicillin-streptomycin (Hyclone). SCC-9 was cultured in DMEM/F12 medium, Cultures were incubated at 37 °C, 5% CO_2_.

### 2.5. Cell Viability Assay

Approximately 5 × 10^3^ cells were seeded on 96-well culture plate (VWR) and incubated for one day. After incubation, the cells were treated with probe CCY-5 for 24 h. To test cell viability in the presence of the probe on cells, we performed cell viability assay using ROCHE Cell Proliferation Kit (Millipore Sigma, Burlington, MA, USA), following the manufacturer’s instructions. The absorbance level was analyzed at 570 nm by Cytation 5 Imaging Reader (BioTek, Winooski, VT, USA). The treated wells relative to that in the control wells and the culture medium was used as a control.

### 2.6. COX-2 Inhibition Assay

Recombinant human COX-2 (1.25 unit) was incubated with 1 μM hematin, 5 mM l-glutatione, 5 mM dopamine haydrochloride, 5 mM EDTA and different concentrations of inhibitors or vector (DMSO) in 150 μL potassium phosphate buffer (pH 8.0) at 37 °C for 15 min. Then 50 μL AA was introduced into the samples at 10 μM to start the reaction. After incubated 10 min, the samples were put on the ice and stopped by adding 50 μL stop solution, which contains 1% formic acid and 100 ng/mL PGE2-*d*_4_ (internal standard). Before using UPLC-MS/MS to measure the PGE2, the pH of samples was adjusted by 0.5 M NaOH solution to around 8.

### 2.7. COX-1 Inhibition Assay

The human COX-1 overexpressed Human Embryonic Kidney (HEK) cells were used to evaluate the COX-1 inhibitory activity of analogs. The cells were cultured in a T-75 flask before using. Then, the cells were washed twice by using PBS and collected by centrifuging at 1500× *g* for 3 min. After discarding the supernatant, the cells were resuspended in 6 mL PBS. Later, the 50 µL cells were incubated with the substrate, AA, with and without inhibitors for 15 min. The amount of 6-keto-PGF1α, metabolites of PGI_2,_ were determined in the presence of different concentrations of inhibitors by LC-MS/MS. SC-560 were used as a positive control.

### 2.8. Fluorescent Microscope Analysis

A total of 2 × 10^4^ cells were seeded on each well of 24-well culture plates (Genesse Scientific, San Diego, CA, USA) and allowed to stabilize for one day. When the cells reached 80% confluency, cells were treated with probe CCY-5 (1 μM in DMSO) at 37 °C in 5% CO_2_ for 30 min. Fluorescence images were visualized by Cytation 5 Imaging Reader (BioTek) with the same condition.

### 2.9. LPS-Induced Inflammation and COX-2 Inhibition and Induction Test

To investigate inflammatory effects, RAW 264.7 was pretreated in DMEM containing 1 μg/mL LPS (O111:B4, Sigma) for 12 h. After pre-incubation in LPS containing media, the cells were washed with PBS three times and treated with 1 μM of probe CCY-5 dissolved in DMSO for 30 min. To test the inhibition effect of COX-2, HeLa cells were incubated with 10 μM of celecoxib for 2 h. After preincubation in COX-2 inhibitor-containing media, the cells were washed with PBS three times and treated with 1 μM of probe CCY-5 dissolved in DMSO for 30 min. The cells were then washed with PBS three times.

### 2.10. Western Blotting Analysis

COX-2 expression at the protein level was confirmed by Western blotting. After RAW 264.7 cells, HeLa cells and SCC-9 cells were stabilized for one day, the cells were collected to extract protein. The protein concentration was determined by Bio spectrometer (Eppendorf, Enfield, CT, USA). Protein samples were separated by sodium dodecyl sulfate poly-acrylamide gel electrophoresis gel (SDS-PAGE,) and electro-transferred onto nitrocellulose blotting membrane (GE Healthcare, Chicago, IL, USA). The membrane was incubated overnight at 4 °C with 1:1000 COX-2 antibody (Cell Signaling, Danvers, MA, USA) and Actin (Invitrogen, Carlsbad, CA, USA) separately. The goat anti-rabbit secondary antibody was used to blot target proteins. Signals were detected by chemiluminescence ECL detection (Advansta, San Jose, CA, USA) and ChemiDoc imaging system (Bio-RAD, Hercules, CA, USA).

### 2.11. Statistical Analysis

Statistical significance of the treatment values was performed by comparing with control values by using Student’s *t*-and indicated as * *p* < 0.05. The data represent the mean ± SE.

## 3. Results and Discussion

### 3.1. Photophysical Properties

The spectroscopic properties of CCY-5 were examined under simulated physiological conditions (phosphate buffer saline, 10 μM, pH 7.4). The probe solution exhibited an absorption peak at 648 nm (Figure 2A) and a strong emission centered at 673 nm (Figure 2B).

### 3.2. The Docking Study of CCY-5 with COX-2

To study the binding action between the CCY-5 and the COX-2, we performed molecular docking using Auto Dock 4.2 (Figure 3). First, celecoxib was redocked into the active site of COX-2. The redocking results showed that the binding pose in the crystal structure was well reproduced where celecoxib formed 2 backbone hydrogen bonds with Leu 338 and Ser 339 and 2 side-chain hydrogen bonds with Gln 178 and Arg 499 (Figure 3B). Such results thus verified that the docking protocol was valid for our ligand–protein system with celecoxib being the ligand and COX-2 being the protein. Next, compound CCY-5 was built in Maestro of the Schrodinger suite. It was then docked into the same active site bound by celecoxib. The results showed that the celecoxib moiety of CCY-5 was located in the same cavity where celecoxib resided when binding alone and the extra fragment was fit into the nearby cavities. In the CCY-5-COX-2 complex, the celecoxib moiety still formed similar interactions with surrounding residues and the extra fragment formed extensive interactions with its surrounding residues such as Ser 516, Asp 348, Leu 103, Tyr 334, Lys 328, and Ile 327 (Figure 3C). These interactions provided a probable binding mode of CCY-5 in the active site of COX-2. These interaction diagrams indicated that different ligands formed different interaction bonds with the COX-2 receptor. This result was consistent with the COX-2 inhibition assay that CCY-5 was a potent inhibitor of COX-2. The inhibition assay showed the IC_50_ value of inhibiting purified COX2 was 0.14 µM for celecoxib and 1.57 µM for CCY-5 (Figure 4A). According to the results of the COX-1 assay, the IC_50_ of CCY-5 against COX-1 is greater than 1mM. As it exhibits good selective COX-2 inhibitory activity (IC_50_ = 1.57 µM), the COX-2 selectivity ratio of CCY-5 is more than 500 (Figure 4B). The tests in vitro encouraged us to explore the cytotoxicity of CCY-5 and pave the way for the next cellular imaging study.

### 3.3. Cell Viability

We tested the cell viability of CCY-5-treated cells to see whether this probe can be used in biological systems. RAW 264.7 (normal cell line) and HeLa cells (cancer cell line) were treated with CCY-5 at various concentrations showing very low cytotoxicity as seen in Figure 4B, which strongly supports the hypothesis that CCY-5 could be used as a fluorescent probe to label cancer cells in biological conditions.

### 3.4. Fluorescent Microscopy Images of Normal and Cancer Cell Lines

In consideration of the promising fluorescent behavior shown by CCY-5, we employed CCY-5 as an imaging agent to detect COX-2 in living cells. A Western blotting assay was first applied to determine the expression of COX-2 in normal cell line, RAW 264.7 cells, and two human cancer cell lines, human squamous cell carcinoma (SCC-9) and human cervical cancer (HeLa) cells. The results showed that COX-2 was highly expressed in both cancer cell lines, but not in the RAW 264.7 cell line (Figure 5A). In our Subsequent study, RAW 264.7, SCC-9 and HeLa cells were incubated with CCY-5 for 30 min. Fluorescence imaging of the cells showed that the SCC-9 and HeLa cells exhibited a stable and robust florescence upon excitation at 628 nm, while the RAW 264.7 cells showed negligible fluorescence towards CCY-5 (Figure 5B and Appendix A). These results indicated that CCY-5 could potentially differentiate among cancer and normal cells. Thus, we presume that the mechanism of the differential luminescence enhancement in the imaging experiments was due to the binding of CCY-5 to COX-2 in the cells. In normal cells, the COX-2 expression was low. Hence, CCY-5 would not accumulate and no luminescence enhancement would be observed in normal cells. On the other hand, cancer cells exhibit an intensive expression of COX-2. CCY-5 was able to bind to COX-2 and is accumulated in the cells, thus generating a high level of luminescence in cancer cells.

### 3.5. Fluorescence Images of LPS-Treated Cells

To gain an insight into the applicability of the CCY-5 to detect intracellular COX-2 in cells induced by cellular oxidative damage, we first prepared RAW 264.7 as control cells. Upon incubation with LPS for 12 h, followed by treatment of CCY-5 for 30 min, we observed an enhanced fluorescence of the probe CCY-5 in the cell lines because the COX-2 level was increased by the LPS-induced oxidative stress (Figure 6A). This result demonstrated that CCY-5 could be used to monitor the oxidative inflammation in the live cellular environment.

In vitro imaging showed that CCY-5 displayed very weak fluorescence in RAW 264.7 cells due to low COX-2 expression. However, a strong fluorescence was detected when the RAW 264.7 cells were pretreated with LPS for 12 h (Figure 6B and Appendix A). The increased expression level of COX-2 in LPS-induced RAW 264.7 cells resulted in enhanced fluorescence. The results suggested that CCY-7 can be used to detect the intracellular COX-2 expression level in LPS-treated RAW 264.7 cells.

### 3.6. COX-2 Inhibitory Effect

Furthermore, to confirm that CCY-5 could specifically target COX-2, the expression of COX-2 was blocked by treatment with celecoxib which specifically inhibits the expression of COX-2. Since the COX-2 enzymes are known to be overexpressed in HeLa cells, and we also previously confirmed that CCY-5 treatment enhances fluorescence in HeLa cells (Figure 7), HeLa cells were chosen as the control cells in this experiment. For the inhibitory studies, HeLa cells were incubated with 10 μM of celecoxib for 2 h, respectively. As seen in Figure 7B and Appendix A, a significant reduction in the fluorescence intensity were observed in HeLa cells upon inhibitor treatment because of the depletion of the intracellular COX-2 levels, prior to the addition of CCY-5, which resulted from the treatment of HeLa cells incubated with inhibitors 2 h. The Western blotting analysis also confirmed that the COX-2 levels are down-regulated in the presence of the inhibitors (Figure 7A). Thus, the celecoxib-guided probe CCY-5 showed a clear dependence on the COX-2 expression levels. Taken together, the evidence shows that the luminescence of CCY-5 is linked to COX-2 expression, suggesting, in turn, that CCY-5 could specifically recognize COX-2 in human cancer cells.

## 4. Conclusions

In conclusion, with the aim of selective bio-imaging of cancer cells that overexpress COX-2, we synthesized and characterized a celecoxib-conjugated fluorescence probe-CCY-5. We observed that its fluorescence intensity in various cells depended on the COX-2 levels by using fluorescence microscopy. Compared with cells that normally express little or no COX-2 (RAW264.7 cells, untreated), the probe showed an increased fluorescence in cancer cells (SCC-9 and HeLa cells), where COX-2 is highly expressed. LPS-treated RAW 264.7 cells with high COX-2 levels also showed enhanced fluorescence. On the other hand, upon co-treatment with a COX-2 inhibitor such as celecoxib, the fluorescence intensity of CCY-5 in HeLa cells was decreased. From these results, we confirmed that the newly synthesized CCY-5 has a remarkable targetability towards cancer cells over normal cells concerning the COX-2 levels, and it can be used as a selective bio-imaging agent for cancer cells. Therefore, CCY-5 could be used as a promising cancer-labeling tool with improved COX2 binding ability and cellular uptake, which is crucial for efficient early cancer diagnosis. To our best knowledge, this is the first work applying CY5-celecoxib conjugate as an imaging agent for COX-2. We demonstrated that CCY-5 differentiates cancer cells from normal cells with high stability and low cytotoxicity.

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
