# Peer review of "Design and Synthesis of a Novel NIR Celecoxib-Based Fluorescent Probe for Cyclooxygenase-2 Targeted Bioimaging in Tumor Cells"

_molecules, 2020, doi:10.3390/molecules25184037_

Round 1

Reviewer 1 Report

The manuscript composed by Hu and co-workers describes the preparation of a novel celecoxib-derived COX-2 fluorescent probe (CCY-5) to use as a monitoring agent in cancer diagnosis and treatments. Computational studies have been performed to define the binding modes of the prepared probe in COX-2, further test of cell viability of CCY-5-treated cells have been reported. Since the fluorescent properties of this novel conjugate, it was investigated as a imaging agent to detect COX-2 in living cells and, importantly, it was found that this probe did not accumulate in normal cells where COX-2 expression is low, whereas in cancer cells it is accumulated and gives high levels of luminescence. The potential use as an imaging agent for COX-2 was here definitely demonstrated. This work displays an interesting and innovativesynthetic aspect that is to highlight: the design of the CCY-5 compound is counter-current, since the attachment of the fluorophore CY-5 is performed on the pyrazole ring side instead of modifying the N-atom in the sulfamide group as traditionallydone. The authors rightly emphasized this risky choice, but unfortunately this topic is not well-described and conveniently supported by experimental characterization in the proper sections. Accordingly, several changes and implementations should be done by the authors before the final acceptance of the manuscript. In details:

1) The preparation of the fluorescent probe has to be moved from Results and Discussion to Experimental Section (from 3.1.1 to 3.1.3). 2) Synthesis of compound 2: please specify the catalytic amount of KI, the extended names of eluents used, PE and EA. Insert the characterization of the compound, especially the IR-spectrum data, particularly characteristic for the azide group. 3) Synthesis of compound 3: please, delete “several drops” and leave the exact amount of 0.5 mL, control the MHz of NMR spectra. (500 or 600?) The proton signal associated to the amine group is not listed, please insert if it is a forgotten.  4) Synthesis of CCY-5: please, insert the reaction yield. 1) For all the synthesized compounds, the complete characterization (proton and carbon NMR, IR, HR-MS spectra) has to be reported as Supplementary Files. I’m convinced the paper would significantly benefit  from this addition. 5) 3.1 Paragraph: the description of the synthetic steps is not satisfactory and it should be accompanied by comments on the spectra. Further, the authors activated the amino-group of compound 3 with EDCl but, to the best of my knowledge, this coupling reagent works better with the -COOH group. So, I advise that the authors could obtain better results changing the order of addition of CY-5 and compound 2. Did the authors just try to make these suggested changes? In my opinion, an attempt should be done and a proper comment on such a synthetic approach should be given in the text, useful for other readers involved in the preparation of similar conjugates. 6) Finally, as a general recommendation the authors should avoid, when possible, abbreviations and, when used, they have to give the extended names at their first appearance. The massive use of abbreviations hampers the reading! (see for example LPS in the abstract or CMP reported in line 79)

Concluding, a huge revision of the paper is strongly recommended by following the Referee’s comments and suggestions. After this operation and positive reconsideration, the manuscript can be published in Molecules journal.

Reviewer 2 Report

The submitted manuscript titled “Design and Synthesis of a Novel NIR Celecoxib Based Fluorescent Probe for Cyclooxygenase-2 Targeted Bioimaging in Tumor Cells” by Ming Hu and co-authors presents a novel COX-2 probe (CCY-5) with high targeting ability and a near-infrared wavelength to probe COX-2 in human cancer cells.. However, the probe reported in the manuscript is developed by attaching an existing COX-2 binding drug to a regular fluorophore. So the novelty of this study is unclear. However the authors present a neat investigation and the manuscript could be published in  the Molecules journal after incorporating the following suggested major revisions.

  1. The authors attach the fluorophore to the CF3 group and not the sulfonyl urea group, what is the rationale behind this probe design and how does it affect specificity of COX-2 binding?
  2. The authors report a 73.4% yield of compound 2, but there are no purification and characterisation details. This needs to be provided if the authors claim this yield.
  3. The authors have not tested the specificity of the probe to COX-2 against other COX proteins. These studies need to be performed to validate the “high targeting ability” as claimed by the authors.
  4. In 3.2.1, the authors discuss the in silico experiments investigating the docking and binding interactions, but have titled the section as binding affinity, this is incorrect.
  5. Also, experiments that measure the binding affinity of the CCY-5 probe for COX-2 and other COX proteins must be performed to understand how well the CCY-5 probes for COX-2. These results must then be compared with the measured binding affinity of celecoxib to obtain a better picture about the efficacy of CCY-5.
  6. Figure 4B suggests CCY-5 is more toxic to RAW 264.7 cells compared to HeLa for 12.5 uM concentration, do the authors suggest any reason for this?
  7. Figures 5C, 6C and 7C are captioned “Quantitative image 285 analysis of the average total fluorescence intensity of ….”. It appears that the studies have only been performed once. These fluorescence microscopy experiments must be performed in triplicates and the error/standard deviation of these results must be reported.
  8. Figure 5B shows the fluorescent and bright field images of different cell lines, however, SCC-9 has a significantly lower cell density compared to that of RAW264.7 and HeLa cells. This should be corrected. Different cell lines must be plated at the same density to avoid the results being skewed due to cell number. This is also the issue with LPS+ cells in Figure 6B.
  9. The authors use 1 ug/mL LPS for studies shown in Figure 6. Is there a reason the authors use such high concentration of LPS? The normal concentrations for LPS stimulation are usually in the 5 – 100 ng/mL range. The authors could perform a study evaluating the effect of different concentrations of LPS on COX-2 in RAW264.7 macrophages.
  10. The caption of figure 6B does not include the concentration of LPS used. All concentrations must be indicated in the captions throughout the manuscript.
  11. Also the macrophage cell line should be referred to as RAW 264.7 and not Raw 264.7.

Round 2

Reviewer 1 Report

I’m satisfied with the changes made by the authors and I think now the manuscript is worthy to be  published after further text editing procedure.

Reviewer 2 Report

The authors have addressed most of the comments outlined below. The authors are, understandably, unable to perform further experiments suggested in 8. and 9. below. While this would help characterise the authors’ probe better, the reviewer agrees that these experiments do not significantly affect the conclusions of the authors’ investigations. Therefore, the revised version of this manuscript is recommended to publication in the Molecules journal, following moderate language corrections.

__________________________________________________________________________________

The submitted manuscript titled “Design and Synthesis of a Novel NIR Celecoxib Based Fluorescent Probe for Cyclooxygenase-2 Targeted Bioimaging in Tumor Cells” by Ming Hu and co-authors presents a novel COX-2 probe (CCY-5) with high targeting ability and a near-infrared wavelength to probe COX-2 in human cancer cells.. However, the probe reported in the manuscript is developed by attaching an existing COX-2 binding drug to a regular fluorophore. So the novelty of this study is unclear. However the authors present a neat investigation and the manuscript could be published in  the Molecules journal after suggested major revisions.

  1. The authors attach the fluorophore to the CF3 and not the sulfonyl urea group, what is the rationale behind this probe design and how does it affect specificity of COX-2 binding?
  2. The authors report a 73.4% yield of compound 2, but there are no purification and characterisation details. This needs to be provided if the authors claim this yield.
  3. The authors have not tested the specificity of the probe to COX-2 against other COX proteins. These studies need to be performed to validate the “high targeting ability” as claimed by the authors.
  4. In 3.2.1, the authors discuss the in silico experiments investigating the docking and binding interactions, but have titled the section as binding affinity, this is incorrect.
  5. Also, experiments that measure the binding affinity of the CCY-5 probe for COX-2 and other COX proteins must be performed to understand how well the CCY-5 probes for COX-2. These results must then be compared with the measured binding affinity of celecoxib to obtain a better picture about the efficacy of CCY-5.
  6. Figure 4B suggests CCY-5 is more toxic to RAW 264.7 cells compared to HeLa for 12.5 uM concentration, do the authors suggest any reason for this?
  7. Figures 5C, 6C and 7C are captioned “Quantitative image 285 analysis of the average total fluorescence intensity of ….”. It appears that the studies have only been performed once. These fluorescence microscopy experiments must be performed in triplicates and the error/standard deviation of these results must be reported.
  8. Figure 5B shows the fluorescent and bright field images of different cell lines, however, SCC-9 has a significantly lower cell density compared to that of RAW264.7 and HeLa cells. This should be corrected. Different cell lines must be plated at the same density to avoid the results being skewed due to cell number. This is also the issue with LPS+ cells in Figure 6B.
  9. The authors use 1 ug/mL LPS for studies shown in Figure 6. Is there a reason the authors use such high concentration of LPS? The normal concentrations for LPS stimulation are usually in the 5 – 100 ng/mL range. The authors could perform a study evaluating the effect of different concentrations of LPS on COX-2 in RAW264.7 macrophages.
  10. The caption of figure 6B does not include the concentration of LPS used. All concentrations must be indicated in the captions throughout the manuscript.
  11. Also the macrophage cell line should be referred to as RAW 264.7 and not Raw 264.7.